# A remote measurement study of PTSD and cannabis use among veterans: Recruitment, retention, and data availability

Daniel Leightley[1]*, Bistra Dilkina[2], Eric R. Pedersen[3], Emily Dworkin[4], Shaddy Saba[5], Esther Howe[6], Praneeth Thota[2], Sriram Nuthi[2], Angeles Sedano[4], Jordan P. Davis[7]

**1** UK Military Research Group, School of Life Course and Population Sciences, King's College London, London, United Kingdom, **2** Viterbi School of Engineering, University of Southern California, Los Angeles, California, United States of America, **3** Department of Psychiatry and Behavioral Sciences, Keck School of Medicine, University of Southern California, Los Angeles, California, United States of America, **4** Department of Psychiatry and Behavioral Sciences, University of Washington, Seattle, Washington, United States of America, **5** Silver School of Social Work, New York University, New York, New York, United States of America, **6** Department of Psychology, University of California Berkley, Berkely, California, United States of America, **7** RAND, Santa Monica, California, United States of America

* daniel.leightley@kcl.ac.uk

## Abstract

With the growing availability of cannabis and increasing public support for legalization, cannabis use disorder (CUD) rates have risen, particularly among veterans, who face disproportionately high rates of co-occurring posttraumatic stress disorder (PTSD) and CUD. Veterans often use cannabis to manage PTSD symptoms, yet research suggests cannabis may exacerbate and maintain these symptoms. Predicting symptom escalation remains challenging due to the complexity of PTSD-CUD interactions and a lack of predictive tools. Advances in mobile and wearable technology provide new opportunities for real-time monitoring, integrating active (self-reported) and passive (sensor-based) data to improve symptom prediction and intervention timing. This longitudinal study examined the feasibility of collecting data among recently discharged veterans with PTSD and cannabis use. The study employed the MAVERICK mobile app, integrating passive data (e.g., heart rate, sleep, activity) and active data (e.g., self-report surveys) over three months. Phase 1 involved beta testing with 20 veterans, assessing feasibility and acceptability, while Phase 2 recruited 75 veterans to evaluate recruitment, retention, and data completeness. Findings indicate high feasibility, with 91.9% of participants providing passive data and a 68% response rate for daily self-report measures. However, data availability varied across measures. These results highlight the potential for integrating passive and active data to improve symptom prediction and early intervention efforts for PTSD and CUD in veterans. Future research should explore long-term engagement strategies and clinical applications of these digital tools to enhance veteran mental health care.

**Data availability statement:** Due to privacy and ethical considerations, the data are not publicly available but can be obtained from John Prindle (jprindle@usc.edu) upon reasonable request.

**Funding:** Work on this article was supported a grant from the National Institute on Drug Abuse (1R21DA051802) to Jordan P. Davis The funders had no role in study design, data collection and analysis, decision to publish, or preparation of the manuscript.

**Competing interests:** Daniel Leightley is a reservist in the UK Armed Forces. This work has been undertaken as part of his civilian employment. This does not alter our adherence to PLOS ONE policies on sharing data and materials.

## Introduction

With the growing availability of cannabis and rising public support for legalization [1,2], cannabis use disorder (CUD) rates have increased in the general population [2,3]. Veterans from conflicts in Iraq and Afghanistan number more than 2 million in the U.S. but are disproportionately affected by CUD, with rates more than doubling in the past decade [4,5]. Posttraumatic stress disorder (PTSD) is the highest co-occurring mental health disorder among veterans diagnosed with CUD, with 29% of patients in the U.S. Veterans Affairs Healthcare System (VA) who have CUD also meet criteria for PTSD [6]. One explanation is veterans with PTSD rely on cannabis to cope with mood and hyperarousal symptoms common to PTSD [7]. Crucially, the risk of symptom escalation for both conditions substantially increases upon return from deployment, often because individuals are transitioning from high stress environments and adapting back to life outside the military [8,9].

Although there have been significant advancements in improving trauma-exposed veterans' access to behavioral health treatments, about half of veterans still fail to seek or engage with health care services [10,11]. There have been ongoing efforts to identify veterans who need support for mental health and substance use difficulties at the time of reintegration. Improving the ability to detect future risk for PTSD and problematic cannabis use after discharge from the military could lead to more effective prevention and intervention efforts [12–14]. The management of PTSD symptomology and, further, problematic substance use requires ongoing, detailed, monitoring and understanding of how symptoms might fluctuate. This poses some challenges, as prior theory and empirical work suggests there is complexity in co-occurring symptomology. According to self-medication and mutual maintenance models, the use of cannabis to self-medicate PTSD symptoms reinforces cannabis use and may lead to cannabis problems, which in turn can maintain symptoms of PTSD [15,16]. Though some veterans believe cannabis use may be a solution for recurring PTSD symptoms, prior research has noted that cannabis use can exacerbate and maintain PTSD symptoms [15,17,18]. Further, while there are valid measurement instruments for screening and assessing current PTSD and CUD, clinicians do not have similar tools for predicting changes in symptomology. Thus, improving our ability to predict clinically significant escalations in cannabis use and PTSD could improve prevention and intervention efforts, but prediction has been, largely, difficult to accomplish.

Smartphones and wearable technology (e.g., Apple Watch, FitBit, Garmin) provide potential opportunities to improve our ability to predict clinically significant escalations in cannabis use and PTSD symptoms. The use of these technologies has increased exponentially over the past decade, providing a source of both active and passive data from veterans. Active data, which is defined as the end user providing data via brief, self-report, survey-based information is the most common way to collect data in studies attempting to assess behavior among veterans. Active data has been used to understand PTSD and problematic cannabis use, but it could also be used to improve prediction of each condition and contribute to scientific understanding of functional associations between them [14]. Passive (automatic) data collection

provides large amounts of data using unobtrusive, built in sensors and metrics that require little to no active input from the end user. Some of the data that these devices collect, passively, may be important for prediction of PTSD and substance use among veterans, including (but not limited to), physiological data such as heart rate and heart rate variability, sleep data (minutes slept, number of awakenings), and activity data which is transmitted through accelerometer sensors such as physical activity. The combination of both active data streams and passive data streams could potentially provide a real time and nuanced view of veterans' physical and behavioral health and functioning, which could inform treatment delivery. It could further be used to predict future changes in health states – for example signals might be identified to predict a relapse in an individual who has not used cannabis for some time or an escalation in PTSD symptoms. This information could initiate the mobilization of a targeted intervention. These methods have yet to be utilized to study co-occurring PTSD and CUD among veterans, and thus their effectiveness, feasibility, and acceptability in this context are unclear.

The present study is a longitudinal study examining the utility of collecting data among recently discharged U.S. veterans with cannabis use and PTSD. The present study also provides an opportunity to explore the recruitment of veterans with PTSD and problematic cannabis use into a complex digital technology study and describe retention rates and adherence to a protocol which includes passive data collection via smartphone and wearable sensors, app-based questionnaires, daily diary sampling method and traditional web-based assessment. Below we aim to 1) summarize study recruitment, retention, and completion rates of data streams (both active and passive); and 2) describe the availability of data from both active and passive data streams. To explore feasibility and acceptability, we also provide some excerpts from our beta test qualitative interviews as well as explore any differences in active or passive data streams based on participants having their own wearable device or one that was provided to them by the study team. Here, we refer to "data availability" as data which is readable and not a duplicate which can be used for analysis.

## Methods

This study received ethical approval from the University of Southern California Institutional Review Board (UP-20–01250). All participants provided informed consent prior to taking part. Recruitment was undertaken between 11-03-2022 and 09-12-2022.

### Study design

All data streams (both active and passive) were collected through the mobile app called MAVERICK using IONIC Capacitor, developed and managed by our team. The app was developed as a Research Viable Product [19] following a co-design with UK military serving and ex-serving personnel, with data privacy at the heart of the development. The MAVERICK app is a custom-made digital platform designed to gather both active and passive data to monitor PTSD symptoms and cannabis use . During the study period, participants completed a brief daily questionnaire via the app, while passive data, such as physical activity, sleep, and other sensor-based metrics, was continuously collected in the background through linkage with Apple Health or Google Fit.

The present study included a two-phase design. The first phase included a split-half beta test design using 20 participants. Here, we beta tested the MAVERICK application and data collection procedure (e.g., active and passive). Phase 1 allowed participants to use the MAVERICK application for 2 weeks and, at the end of the best test period, conducted a brief qualitative interview with each participant to explore acceptability and feasibility. The second phase used information from phase 1 to improve MAVERICK usability and acceptability and to recruit 74 recently discharged veterans with elevated PTSD symptoms and problematic cannabis use via social media platforms. To test feasibility of recruitment and retention in phase 2, half of recruited participants were required to own a wearable device (e.g., Fitbit, Garmin Watch, Apple Watch) with the remaining half being provided a FitBit device. This allowed our team to test a potential cost-saving measure (including only participants who own a wearable device) as well as determine the feasibility of providing wearable devices to participants.



Each participant was asked to wear the FitBit or other wearable device for 3 months, and as much as possible throughout the day and night. Participants also completed a screening survey at baseline, three follow-up surveys (1-, 2-, and 3-months post-baseline; with each month representing a 28-day period), and daily data for 3 months following their baseline survey. Participants were asked to complete daily data twice per day (morning and evening) with passive data being collected via the wearable device and MAVERICK continuously for the study duration. Participants were compensated for monthly assessments as well as daily assessments, and those who were provided a Fitbit were able to keep it.

## Study population

Eligible participants were recruited via several channels including using the recruitment services of BuildClinical. BuildClinical is a technology company that enables academic researchers to engage the exact populations needed for a given study using online and social media-based advertisements. BuildClinical uses machine-learning and data mining to identify specific patient groups across a multitude of platforms (e.g., social media, health websites, etc.). Participants responding to online ads were contacted by phone by a research assistant who confirmed their eligibility and used use a series of additional validation checks to limit fraudulent participants (e.g., ensuring phone responses match survey responses to items such as military branch and rank. All participants provided written consent, and all procedures were approved by the University of Southern California Institutional Review Board (IRB #UP-20–00853). Eligibility criteria for the beta test (phase 1) and the pilot study (phase 2) were: (a) veteran aged 18 or older separated or discharged from military service from the Air Force, Army, Marine Corps, or Navy within the past 3 years; (b) not currently affiliated with active duty service or in the reserves or guard units; (c) served as part of OEF/OIF or Operation New Dawn (i.e., Iraq/Afghanistan); (d) able to read English; (e) own a personal smartphone released since 2012 with Internet access and have interest in using apps on that phone; (f) not receiving treatment for cannabis, alcohol, or other drug use or PTSD at the VA or other health care provider; (g) past-month use of cannabis, (h) trauma exposed during military service, and (i) with a PC-PTSD score of 1 or more associated with military trauma exposure, representing at least some symptoms of PTSD

## Data collection

**Active data collection.** The active component collected a range of self-report measures, via a visual user interface on the MAVERICK app. Self-report measures (see Table 1) were collected at the following intervals: 1) at baseline, 2) morning (between 6am and 11.59am) and evening (between 6 pm and 11.59 pm) and 3) monthly follow-up (1-, 2-, and 3- month follow-up). At baseline, key demographics were collected from each user. To encourage users to complete the self-report questionnaire, in-app notifications were sent via the smartphone notification system. All user interactions with MAVERICK were recorded to analyze usability and acceptability.

**Passive data collection.** The passive component of MAVERICK collects physical activity information from a number of sensors and data sources (e.g., Apple Health, Google Fit, Garmin, GPS location) and streams the information to a cloud-based infrastructure on a daily basis. This approach enabled participants to use a range of different devices and smart watches and the framework aggregated this data into a single format for use. In this study, each participant was asked to link their wearable device to the MAVERICK app. This allowed for accurate recording of steps, heart rate, sleep, and heart rate variability and other data streams. Written and audio-visual instructions were provided detailing the procedure for linking the app to the wearable device. To reduce impact on users' devices, this information is collected at a minimum of 30-minute intervals. All data extracted was time-encoded and stored separately from user identifiable information.

## Analytic plan

First, we provide some brief excerpts from our phase 1 qualitative interviews that explored feasibility and acceptability. We asked participants about acceptability and usability of the MAVERICK application, acceptability of collecting passive

**Table 1. Baseline demographics.**

| Variable | M(SD) or N(%) |
|---|---|
| Age | 34.2 (8.01) |
| Female | 14 (18.9%) |
| **Race/ethnicity** | |
| White | 34 (45.9%) |
| Hispanic | 27 (36.5%) |
| Black | 7 (9.5%) |
| Asian | 2 (2.7%) |
| American Indian/Alaska Native | 1 (1.4%) |
| Other/Multi-race | 18 (24.3%) |
| Num. of deployments | 2.49 (2.91) |
| **Employment** | |
| Part-time | 20 (27.0%) |
| Full-time | 25 (33.8%) |
| Unemployed | 17 (23.0%) |
| Retired | 12 (16.2%) |
| PTSD | 39.0 (16.3) |
| Depression | 10.4 (5.66) |
| **Substance use** | |
| Cannabis (days/month) | 20.4 (11.1) |
| Cannabis consequences | 3.23 (3.63) |
| Alcohol (days/month) | 8.71 (8.18) |
| Binge (days/month) | 3.38 (4.84) |
| Study provided watch (FitBit) | 32 (42.1) |
| Own an Apple smartphone | 54 (76.1) |

data streams, and hypothetical questions regarding the acceptability of sharing data collected with either VA or non-VA organizations.

For phase 2 data, we provide simple baseline characteristics of the sample using means and standard deviations or N and percent. The number and percentage of people who have provided any data via the MAVERICK app and the wearable device throughout the course of follow-up have been summarized, then divided into quartiles to examine the numbers of people who have provided 0–25% of expected data, 26–50%, 51–75 and>75% of data throughout the study. This has been stratified by wearable ownership (i.e., was a participant supplied with a wearable or brought their own).

For the analysis of data availability and participant engagement with the app, descriptive and inferential statistical methods were used. Active data collection was presented using the median and interquartile range (IQR) to summarize the number of completed questionnaires per participant. The total number of questionnaires collected, and the standard deviation (SD) were also reported to provide an overall measure of variability. For passive data collection, we analyzed the proportion of participants providing passive data and the average number of days data was collected per participant, along with the associated standard deviation. The total volume of compressed data collected was reported based on the number of data rows. Data availability was visualized using stratified plots by participant and study day to observe trends over time. App usage analytics were reported using descriptive statistics, including the median, IQR, and standard deviation to summarize participant engagement with the MAVERICK app. This data was extracted from Google Analytics and server logs.



## Results

### Phase 1: Qualitative interview responses regarding acceptability and feasibility

Following the beta testing phase in which 20 veterans used the MAVERICK application for 2-weeks, we conducted brief, qualitative interviews with each beta-test participant. Initially, we asked each participant about the acceptability and usability of the MAVERICK application data collection process. Participants, in general, reported high usability and ease of use with one participant noting: "*I felt like it was easy to use, all the buttons worked. I went to the settings a couple of times and didn't have any issues with the settings. So I would say ease of operation and like the layout of it was pretty self –explanatory*" Participants generally appreciated receiving reminders for doing the daily surveys, however, some participants felt the daily surveys made the questions seem repetitive, and reported challenges with recall for the length of time they had to remember (e.g., 8 hour windows). However, some participants did mention that completing the daily surveys allowed them to develop more self-reflection and awareness. For example, one participant stated

"*You know, I like to, I like to smoke cannabis and I smoke cannabis like pretty much daily, but I'm, I'm thinking about thinking about actually maybe quitting or, or going on a tolerance break because it, it really, it doesn't do anything at all to me anymore. It just kind of relaxes me, but, uh, it doesn't, doesn't really do a whole lot anymore.* "

However, not all participants experienced this, with some noting the questions were emotionally difficult. For example, one participant stated:

"[*the questions*] *that were asking about like, uh stressful experiences about, uh like, military service, or when asking to think back on a stressful experience and if when it asks you do you blame yourself or someone else or, um what else, do you feel guilty or shame that, type of stuff. It kinda really had me reflect.*"

We also asked participants about how they felt and how they think their fellow veterans would feel regarding collecting the passive data from their phones and wearable devices. For the most part, participants had mixed (both positive and negative) feelings about these data being collected. For example, one participant stated "*I'm torn. If it helps, you know, it's definitely a great thing. It's just, it sucks that it's so intrusive.*" Another participant had a different perspective noting that the data are being collected anyway by other companies stating:

"*I understand that, you know, all that data is out there and that information is, it's already being collected and it's out there somewhere and somebody has it. I feel like it's being used to benefit myself and or someone else in some way, I'm happy for it to go out there. It's not attributable as long as it's unattributable to me or at least you know, to like my social security number. I don't have a problem with it. I can't think of a problem that I have with you guys collecting the data that you asked for that health data because it only paints a more complete picture.*"

Finally, we asked participants how they or their fellow veterans might feel if we proposed a project in which these data were shared with the VA or other organization. Here, responses were much more negative and resistant to the idea. For example, one veteran stated

"*I don't see it being a big hit among the veteran population. There is a lot of the old way of thinking and, you know, some of my friends don't even know what data is. So the guys that I served with and had to explain it to a couple of guys, like, you know, every time you click the phone or move the phone or go anywhere, they are tracking every little bit of information they can. But, so with that, the VA could potentially use that information, all that data, in ways to maybe avoid someone's disability claim.*"

However, some veterans were in favor of their care team at the VA or other health care organization having these data. Most of the positive response to these questions was in relation to wanting to receive the best and most useful care for their concerns. For example, one veteran stated:

*"Of course, I'm concerned with privacy. The problem with that is that it's an outsized concern in my case because in my case I want to be treated for what's wrong with me. And in my case, I know the more information that my care team has, the better the care and more complete and more holistic that care is going to be. So I have to be willing to give that privacy part up to get something in return"*

### Phase 2: Recruitment and retention rates

Fig 1 shows a detailed flow chart of participant recruitment and reasons for non-participation. In terms of retention during the three-month study period, 92% completed the 1 month follow-up, 76% completed the 2 month follow-up, and 72% completed the 3 month follow-up (see Fig 1). In terms of daily diary collection (note: participants were asked to fill out one survey in the morning and one in the evening), on average the number of days with any daily data was 57 (out of 84 total) with an average response rate of 68%. Fig 1 also depicts daily diary retention broken down by each month of the study period.

### Phase 2: Sample characteristics

Sample characteristics can be found in Table 1. Participants were, on average, 34 years old with 18.9% (n = 14) being female, and most owned a Apple smartphone (n = 54). Most participants identified as White (45.9%) with 36.5% identifying as Hispanic, 9.5% as Black, 2.7% Asian, 1.4% American Indian/Alaska Native, and 24.3% Other/multi-race. Veterans in this sample had an average PTSD score of 39.0 with 59.5% meeting the cutoff for probable PTSD. Veterans also reported using cannabis on an average of 20 days in the past month with an average of 3.23 cannabis related consequences in the past month.

### Phase 2: Data availability

**Active data collection.** Table 2 provides a comprehensive list of measures that are available from active data streams. The table includes brief descriptions of each measure or item, and when it was utilized (monthly assessments, daily assessments). Overall, participants completed a median of 106 (IQR:45–152) daily questionnaires, with a total of 7,310 (SD: 59.5) questionnaires provided in total. For AM surveys, participants completed a median of 61 (IQR:22–76) questionnaires, with a total of 3,334 (SD: 28.7) questionnaires. For PM surveys, participants completed a median of 58 (IQR:26.3 to 78) questionnaires, with a total of 3,739 (SD: 29) questionnaires. For monthly follow up surveys, participants completed a median of 3 (IQR:1–3) questionnaires, with a total of 165 (SD: 1.21) questionnaires.

**Passive data collection.** A total of 68 (91.9%) participants provided data across the study period, with the average number of data days per participant in which data was collected was 49.3 (SD: 12.5). A total of 126 gigabytes of compressed data was collected which represents raw compressed data across all participants, including wearable sensor streams and app-generated logs. Fig 2 presents passive data availability stratified by participant and study day. It was observed that passive data availability decreased over time.

Table 3 presents wearable passive data availability (more details of all passive data collected can be found in supplemental materials, item A). Data collection relied on 1) participants wearing the wearable device, 2) regularly charging and syncing the wearable device; and 3) using the MAVERICK app. For steps data, the majority (66.2%) achieved a completion rate of 75% or higher. SPO2 data showed a large portion (48.6%) with 0–25% completion, while only 16.2% had 75%+ completion. Sleep data exhibited 60.8% in the 26–49% range, with a smaller proportion (10.8%) achieving

**Not assessed for Eligibility**
**(N =485; 38.9%)**
- 174 Not contacted because enrollment goal was met
- 311 New leads/ Ineligible – not contacted
  - ○ 9 No PTSD endorsed
  - ○ 17 No cannabis use
  - ○ 6 Active in treatment
  - ○ 1 Not a US Veteran
  - ○ 278 Multiple reasons

**Total Contacts**
**N = 1,248**

**Not Eligible**
**(N = 687; 90.0%)**
- 5 Did not serve in the US Military
- 17 Did not endorse PTSD
- 41 No cannabis use
- 248 Not discharged within the past 5 years
- 2 Did not have a smart phone
  19 Active in treatment
- 4 No longer interested
- 7 fake accounts
- 4 enrolled in another study
- 2 coast guard
- 38 screened/Didn't follow through
- 159 contacted and did not respond
- 141 Multiple reasons

**Total willing and assessed for eligibility**
**N = 763**

**Total eligible/enrolled**
**N = 76**

**1 Month**
**N = 62 (92%)**

**Daily surveys 1 month**
- Avg days with daily data: 22
- % days with any daily data: 77%

**2 Month**
**N = 53 (76%)**

**Daily surveys 2 month**
- Avg days with daily data: 18
- % days with any daily data: 65%

**3 Month**
**N = 53 (72%)**

**Daily surveys 3 month**
- Avg days with daily data: 17
- % days with any daily data: 60%

**Fig 1. STROBE flowchart for recruitment and retention.**

**Table 2. Measures available from active data streams.**

| Domain | Measure | Baseline and monthly | Daily |
|---|---|---|---|
| PTSD symptoms | The PTSD Checklist for DSM-5-Military Version (PCL-5) was used to PTSD symptoms at baseline. This measure has been validated with veterans. The 4-item PCL was used during daily data collection | x | x |
| Problematic cannabis use | The Cannabis Use Disorder Identification Test-Revised is an 8-item questionnaire that assesses problematic cannabis use and probable CUD in the past 6 months. | x | |
| Cannabis use and related problems | Cannabis use was measured using an online version of the timeline follow back to establish a retrospective account of the number of days and times participants used cannabis. Participants estimate the typical number of times they used cannabis on each day of the week during the past month. Number of problems associated with cannabis use was assessed with the Brief Marijuana consequences questionnaire, which measures the number of cannabis problems during the past 30 days. | x | |
| Daily Cannabis use | Daily cannabis use (morning and evening) will be measured using three items that assess daily use of cannabis. Each participant will be asked "have you used cannabis since your last assessment?", "how many hours have you been high today?", "at what time did you first use cannabis today?" and "at what time did you last use cannabis today?" | | x |
| Cannabis motives | Motives for use was measured using the marijuana motives questionnaire. | x | |
| Life events checklist | The Life Events Checklist (LEC-5) is a self-report measure designed to screen for potentially traumatic events in a respondent's lifetime. The LEC-5 assesses exposure to 16 events known to potentially result in PTSD or distress and includes one additional item assessing any other extraordinarily stressful event not captured in the first 16 items. | x | |
| Insomnia severity | Insomnia severity was assessed using the 7-item Insomnia Severity Index. A score of 10 or more indicates possible insomnia diagnosis. | x | |
| Daily Sleep | The number of hours of sleep (entered as number of hours) and sleep quality (rated from 1 to 5) was used to assess sleep quality among veterans. | | x |
| Social support | Multidimensional scale of perceived social support | x | x |
| Impulsivity | Impulsivity was assessed using the short UPPS scale. This scale has 20 items that break into 5 dimensions: lack of premeditation, lack of perseverance, sensation seeking, positive urgency, and negative urgency. | x | |
| Positive and negative affect | The 20-item positive and negative affect scale (PANAS) scale. A short, 10 item, version was used for daily data assessments | x | x |
| Pain | Pain intensity (rated from 0–10) was assessed each day, Pain interference using the PROMIS pain interference scale and Pain anxiety using the Pain Anxiety Symptom scale were assessed at the monthly intervals | x | |
| Daily pain | A 0–10 numerical rating scale of pain intensity was used at the daily level | | x |
| Stress | The 10-item perceived stress scale was used for monthly assessments | x | |
| Daily Stress | A slider from 1 (lowest) to 5 (highest) was used to assess stress at the daily level | | x |
| Depression | The PHQ8 was used to assess depression during the monthly assessments | x | |
| Anxiety | The GAD7 was used to assess generalized anxiety disorder symptoms | x | |
| Daily depression and anxiety | Two sliders from 0–5 were used to assess depression and anxiety with higher scores indicating higher depression or anxiety | | x |
| Loneliness | Loneliness was assessed using the UCLA loneliness scale | x | |

75%+completion. Respiration rate data indicated 59.5% in the 0–25% range, and only 9.5% had 75%+completion. Heart rate data was more evenly distributed, with the highest proportion (31.1%) in the 75%+range. Exercise data revealed 51.4% with 0–25% completion, and 20.3% with 75%+completion. Distance data followed a pattern similar to steps, with 66.2% in the 75%+range. Activity data revealed 40.5% with 0–25% completion and 33.8% with 75%+completion. Overall, steps and distance data had the highest completion rates (75%+). Other data types showed varied completion rates with notable portions in the lower ranges (0–25%).

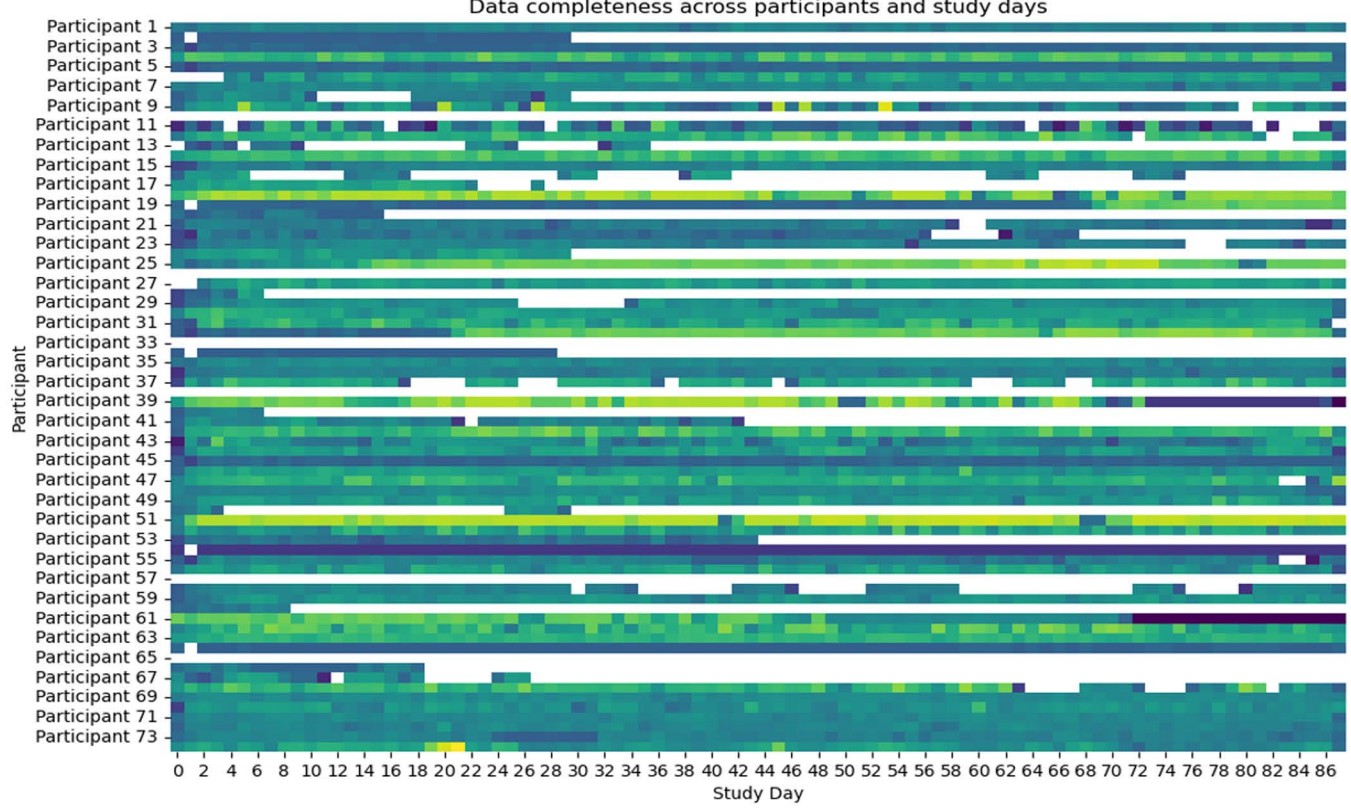

**Fig 2. Data availability across the study period.**

## App usage analytics

Table 4 presents a summary of participant engagement with the MAVERICK app, including interaction metrics, sessions, and interactions with push notifications. The median number of app initializations per participant was 141, with an inter-quartile range (IQR) of 61.3 to 182.8. Participants had a median session count of 513.5, with a wide IQR of 5.0 to 112.0, and a median session duration of 141.7 minutes, with an IQR ranging from 115.5 to 205.1 seconds. Server interactions showed a median of 386.0 per participant, with an IQR of 122.0 to 541.0. In terms of push notifications, participants received a median of 344.5 notifications, with an IQR of 133.0 to 406.5, of which the median number opened was 50.5, with an IQR of 17.5 to 99.0, and the median number dismissed was 34.0, with an IQR of 5.0 to 112.0. Thus, these results indicate significant variability in participant engagement and interaction with the app, as evidenced by the wide interquartile ranges across various metrics, reflecting differing levels of user interaction and responsiveness to app-generated push notifications among the participants.

## Discussion

U.S. veterans are at exceedingly high risk of co-occurring behavioral health problems, with CUD and PTSD being two problems that are especially prevalent [6]. Given that many U.S. veterans have returned from post-9/11 conflicts in recent years, it is imperative that we adequately assess and address risk for these problems in this population. Recent advances in remote measurement methodologies, namely symptom tracking apps and wearable devices that can collect a broad variety of real-time information on individuals' health and functioning, show promise for improving behavioral health

**Table 3. Passive data availability for each data group stratified by watch ownership.**

| Data type | Data availability | Overall N (%) | Study provided watch N (%) | Participant own watch N (%) |
|---|---|---|---|---|
| **Steps** – Tracks step count data collected from various sensors and logs. | 0-25%<br>26-49%<br>50-74%<br>75%+ | 15 (20.3)<br>7 (9.5)<br>3 (4.1)<br>49 (66.2) | 10 (4.8)<br>3 (1.4)<br>3 (1.4)<br>32 (66.7) | 5 (1.3)<br>4 (1.0)<br>0 (0.0)<br>17 (65.4) |
| **SPO2** – Blood oxygen saturation level, recorded as part of health monitoring. | 0-25%<br>26-49%<br>50-74%<br>75%+ | 36 (48.6)<br>16 (21.6)<br>10 (13.5)<br>12 (16.2) | 30 (52.6)<br>12 (21.1)<br>7 (12.3)<br>8 (14.1) | 6 (28.6)<br>4 (19.1)<br>7 (33.3)<br>4 (19.1) |
| **Sleep** – Sleep-related metrics such as time asleep, time in bed, sleep efficiency, and stages (REM, deep, and light sleep). | 0-25%<br>26-49%<br>50-74%<br>75%+ | 18 (21.6)<br>45 (60.8)<br>3 (4.1)<br>8 (10.8) | 12 (23.5)<br>32 (62.8)<br>2 (3.9)<br>5 (9.8) | 6 (26.1)<br>13 (56.5)<br>1 (4.4)<br>3 (13.1) |
| **Respiration Rate** – Includes breathing rate during different stages of sleep (deep, light, and REM) and overall respiratory rate summaries. | 0-25%<br>26-49%<br>50-74%<br>75%+ | 44 (59.5)<br>20 (27.0)<br>3 (4.1)<br>7 (9.5) | 35 (61.4)<br>17 (29.8)<br>1 (1.8)<br>4 (7.0) | 9 (52.9)<br>3 (17.6)<br>2 (11.8)<br>3 (17.6) |
| **Heart Rate** – Includes heart rate data, including resting heart rate, heart rate variability (HRV), and heart rate while in active zones. | 0-25%<br>26-49%<br>50-74%<br>75%+ | 20 (27.0)<br>13 (17.6)<br>18 (24.3)<br>23 (31.1) | 12 (29.3)<br>6 (14.6)<br>9 (22.0)<br>14 (34.2) | 8 (24.4)<br>7 (21.2)<br>9 (27.3)<br>9 (27.3) |
| **Food** – Includes dietary intake and nutritional data. | 0-25%<br>26-49%<br>50-74%<br>75%+ | 34 (45.9)<br>21 (28.4)<br>3 (4.1)<br>16 (21.6) | 11 (40.7)<br>8 (29.6)<br>1 (3.7)<br>7 (25.9) | 23 (48.9)<br>13 (27.7)<br>2 (4.3)<br>9 (19.2) |
| **Exercise** – Includes physical exercises, such as activity name, duration, distance, pace, and heart rate during exercise. | 0-25%<br>26-49%<br>50-74%<br>75%+ | 38 (51.4)<br>9 (12.2)<br>12 (16.2)<br>15 (20.3) | 22 (52.4)<br>6 (14.3)<br>6 (14.3)<br>8 (19.1) | 16 (50.0)<br>3 (9.4)<br>6 (18.8)<br>7 (21.9) |
| **Distance** – Includes data on distances covered, either from daily activities, exercises, or specific activity logs. | 0-25%<br>26-49%<br>50-74%<br>75%+ | 16 (21.6)<br>6 (8.1)<br>3 (4.1)<br>49 (66.2) | 6 (14.3)<br>2 (4.8)<br>2 (4.8)<br>32 (76.2) | 10 (31.3)<br>4 (12.5)<br>1 (3.1)<br>17 (53.1) |
| **Activity** – Including metrics on floors climbed, activity calories, sedentary minutes, and time spent in light, moderate, or vigorous activity. | 0-25%<br>26-49%<br>50-74%<br>75%+ | 30 (40.5)<br>9 (12.2)<br>10 (13.5)<br>25 (33.8) | 17 (30.5)<br>6 (14.3)<br>5 (11.9)<br>14 (33.3) | 13 (40.6)<br>3 (9.4)<br>5 (15.6)<br>11 (34.4) |

**Table 4. Engagement, app interactions and notifications over the study period per participant.**

| Interactions | Medium (IQR) |
|---|---|
| **Engagement measures** | |
| Initializations | 141 (61.3 to 182.8) |
| Session Count | 513.5 (5.0 to 112.0) |
| Session duration | 141.7 (115.5 to 205.1) |
| Server interactions | 386.0 (122.0 to 541.0) |
| **Push notifications** | |
| Received | 344.5 (133.0 to 406.5) |
| Opened | 50.5 (17.5 to 99.0) |
| Dismissed | 34.0 (5.0 to 112.0) |

research and practice and specifically for monitoring symptom change and potentially preventing adverse outcomes [20]. This paper details preliminary work from a pilot study exploring remote measurement of co-occurring PTSD and cannabis use among veterans utilizing both active (self-report via a symptom tracking app) and passive (via wearable devices) data. Below we report lessons learned from our experiences with recruitment and retention of veterans in this study and discuss the potential clinical and research implications of our data.

Considering usage and completion metrics for daily surveys and quantity of passive data collected, we demonstrate a substantial degree of feasibility for remote measurement in our veteran sample. There is some concern among health scientists about whether daily diary studies are overly burdensome and time consuming for participants [21]; our survey protocol for example involved frequent (daily) data collection over a several month period. Encouragingly, participants provided data for nearly 70% of days on average. While participant attrition over time is a common challenge in longitudinal digital health studies, our retention rate of 72% at 3 months is notably high for this type of research. This strong retention supports the feasibility of longer-term remote data collection in veteran populations and provides a sufficiently robust dataset for longitudinal modelling including machine learning. Nevertheless, we acknowledge the potential for bias introduced by attrition and have considered its implications in our interpretation of the findings.

Qualitative data also reflect the app's general acceptability, with participants reporting a high degree of usability. Some participants even noted that responding to daily surveys enhanced their awareness of their behavioral health; however, others found that daily surveys became repetitive or emotionally taxing. But it important to consider that this study may have been subject to selection bias, as participants were required to engage with MAVERICK (a digital platform), potentially favoring those who are more technologically literate. The short follow-up period (3 months) limits insights into longer-term engagement and outcomes. As such, findings may not be fully generalizable to the broader veteran population, particularly those less comfortable with digital tools.

Still, this active data can greatly enhance knowledge of the etiology and symptom presentation of co-occurring PTSD, CUD, and their correlates. As symptoms of these conditions are known to fluctuate daily, this type of data coupled with intensive longitudinal analytic methods can help clarify how symptoms change and influence one another at the within-person level, providing novel information about how short-term changes in one condition or mechanism might affect another condition they have. Several studies have used similar types of data (e.g., passive, or intensive longitudinal data) to explore meaningful associations and mechanism of change [22,23]. Given findings from prior simulation studies of daily diary data [24] and our success in collecting this data, we are well-powered to advance etiological and theoretical knowledge of co-occurring PTSD and CUD among our veteran sample using this rich self-report data. This provides the opportunity for the use of machine learning to help identify symptom escalation and de-escalation as part of a wider clinical intervention.

Integrating active and passive data sources may also enable the development of real-time just-in-time adaptive interventions supported by machine learning. For example, combining daily self-reported mood and/or symptom ratings with physiological data such as heart rate variability or sleep disruption could be used to detect patterns associated with symptom escalation and de-escalation. These integrated data streams could feed into personalized alert systems that notify clinical teams, or even participants themselves, when changes arise, offering an opportunity for timely, just-in-time interventions. To implement these interventions in clinical settings, steps may include integrating wearable data into Electronic Healthcare Records, developing clinician-facing dashboards to visualize trends over time, and for triaging alerts to appropriate care providers. While further validation is required, the integration of these systems could support more proactive, personalized care for veterans with co-occurring PTSD and cannabis use disorder.

Passive data completion and feedback also demonstrates the potential use of wearable devices for addressing co-occurring PTSD and CUD, along with a need to address implementation challenges. Data were collected from wearable devices for over 90% of participants. Some data streams, such as steps and distance traveled, had relatively high rates of completion (with most participants providing data on 75% or more of days). Other variables, such as sleep and respiration,

had relatively sparse completion. As we aggregated data across several devices using device developer tools (e.g., Apple Health or FitBit), discrepancies in completion across some of these variables could be explained by the idiosyncrasies of specific data collection and processing procedures for each variable and device. Thus, while existing software and hardware may provide behavioral health researchers and clinicians with a valuable and accessible toolkit, it may be beneficial to clarify, adapt, and test processes during deployment. High missingness in sleep data may also be due to participants removing their devices before sleep or incorrectly wearing the device, thus, if sleep data is potentially important it is necessary to consider how to ensure devices are worn or it is otherwise collected.

Veterans' qualitative interview responses also convey their mixed perspectives on the acceptability of passive data collection and its clinical applications. Some felt that collecting data of this type was somewhat intrusive, which is understandable given the potential sensitivity of health data collected and prior research demonstrating veterans in particular might have privacy concerns with health technology [25]. Others felt that since this type of data is already being collected, it is worth leveraging it to better understand and help veterans. Some veterans were particularly weary, however, of the idea of health systems like the VA collecting or obtaining this type of data. They cited concerns about the VA denying disability claims on the basis of this data, which again is understandable given many veterans' challenges with obtaining disability benefits [26].

With many veterans in the sample providing a substantial amount of data, and qualitative participants expressing a cautious optimism about the clinical use of this data, our experiences with recruitment and retention in this study suggest it is worth continuing to explore these methods and their applications with veterans who do deem them acceptable in real world settings. Results from subsequent papers with this data will clarify whether valid and accurate predictive model development is feasible with our wearable device data collection protocol, and whether a specific subset of data streams or engaging participants for shorter periods of time would have been sufficient while lowering participant burden. Future work should also explore the feasibility of deploying already validated predictive models "in the field" – and whether, for example, participants are open wearing their wearable devices consistently enough to predict changes in PTSD and CUD before they happen. This may require additional public health education and developing patient-facing agreements and broader system-level protocols to assuage privacy concerns. Continued development of these data collection tools may ultimately pay large dividends in allowing health systems to monitor and predict veterans' behavioral health symptomology, and to provide interventions and prevention strategies that are time sensitive and person-centered [27].

## Supporting information

**S1 File. Data features and aggregation.**
(DOCX)

## Author contributions

**Conceptualization:** Daniel Leightley, Bistra Dilkina, Eric R. Pedersen, Jordan P. Davis.

**Data curation:** Daniel Leightley, Bistra Dilkina, Eric R. Pedersen, Praneeth Thota, Sriram Nuthi, Jordan P. Davis.

**Formal analysis:** Daniel Leightley, Bistra Dilkina, Eric R. Pedersen, Emily Dworkin, Shaddy Saba, Jordan P. Davis.

**Funding acquisition:** Daniel Leightley, Bistra Dilkina, Eric R. Pedersen, Emily Dworkin, Jordan P. Davis.

**Investigation:** Bistra Dilkina, Emily Dworkin.

**Methodology:** Daniel Leightley, Bistra Dilkina, Esther Howe, Jordan P. Davis.

**Project administration:** Angeles Sedano, Jordan P. Davis.

**Software:** Daniel Leightley, Bistra Dilkina, Praneeth Thota, Sriram Nuthi, Jordan P. Davis.

**Supervision:** Daniel Leightley, Bistra Dilkina, Angeles Sedano, Jordan P. Davis.

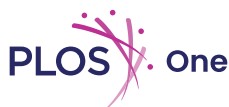

**Validation:** Bistra Dilkina, Jordan P. Davis.

**Visualization:** Daniel Leightley, Eric R. Pedersen.

**Writing – original draft:** Daniel Leightley, Eric R. Pedersen, Emily Dworkin, Jordan P. Davis.

**Writing – review & editing:** Daniel Leightley, Eric R. Pedersen, Emily Dworkin, Shaddy Saba, Esther Howe, Angeles Sedano, Jordan P. Davis.

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
