## [Decision Letter · Decision Letter 0]

1 Aug 2025

Dear Dr. Leightley,

Thank you for submitting your manuscript to PLOS ONE. After careful consideration, we feel that it has merit but does not fully meet PLOS ONE’s publication criteria as it currently stands. Therefore, we invite you to submit a revised version of the manuscript that addresses the points raised during the review process.

We look forward to receiving your revised manuscript.

Kind regards,

Firas H Kobeissy, PhD

Academic Editor

PLOS ONE

Journal Requirements:

2.Thank you for stating the following financial disclosure: [Work on this article was supported a grant from the National Institute on Drug Abuse (1R21DA051802) to Jordan P. Davis]. 

3. Thank you for stating the following in the Competing Interests section: [Daniel Leightley is a reservist in the UK Armed Forces. This work has been undertaken as part of his civilian employment.].

5.Please include captions for your Supporting Information files at the end of your manuscript, and update any in-text citations to match accordingly. Please see our Supporting Information guidelines for more information: http://journals.plos.org/plosone/s/supporting-information.

Reviewers' comments:

Reviewer's Responses to Questions

**Comments to the Author**

1. Is the manuscript technically sound, and do the data support the conclusions?

Reviewer #1: Yes

Reviewer #2: Yes

Reviewer #3: Yes

Reviewer #4: Yes

2. Has the statistical analysis been performed appropriately and rigorously?

Reviewer #1: Yes

Reviewer #2: Yes

Reviewer #3: Yes

Reviewer #4: Yes

3. Have the authors made all data underlying the findings in their manuscript fully available?

Reviewer #1: No

Reviewer #2: Yes

Reviewer #3: Yes

Reviewer #4: Yes

4. Is the manuscript presented in an intelligible fashion and written in standard English?

Reviewer #1: Yes

Reviewer #2: Yes

Reviewer #3: Yes

Reviewer #4: Yes

Reviewer #1: This addresses a critical gap in leveraging digital tools to monitor PTSD and cannabis use among veterans. Strengths include a mixed-methods approach (qualitative insights complementing quantitative data) and rigorous reporting of recruitment/retention metrics. Below are suggestions for improvement:

- Clarify Data Availability: Ensure supplementary materials include aggregated data (e.g., summary tables) if raw data cannot be shared. Specify whether de-identified datasets will be archived in a repository post-publication.

- Expand Limitations: Discuss potential selection bias (e.g., tech-savvy participants) and the short follow-up period (3 months). Acknowledge that feasibility in this sample may not generalize to broader veteran populations.

- Strengthen Discussion: Elaborate on how passive/active data integration can inform clinical interventions (e.g., real-time alerts for symptom escalation). Highlight practical steps for implementing these tools in care settings.

- Proofreading: Address minor grammatical errors and simplify complex sentences (e.g., in the Abstract and Introduction).

Reviewer #2: Quite an interesting topic and a unique demography considering the rate at which PTSD impacts the veterans, albeit age or length of service. PTSD has been linked to several other neurodegenerative and cardiovascular diseases. This article is novel and could yet help find sustainable means of managing PTSD among veterans.

Reviewer #3: Consider including more details about the machine learning techniques or predictive models used (or planned for future use) to analyze the collected data, as this is mentioned in the abstract but not elaborated upon in the methods or results sections.

The recruitment flowchart (Figure 1) is informative, but the reasons for ineligibility (e.g., "Multiple reasons") could be further clarified to enhance transparency.

The retention rates are commendable, but the manuscript should address potential biases introduced by attrition, particularly in the later months of the study (e.g., 72% retention at 3 months).

The variability in passive data completion rates (e.g., high for steps/distance but low for SPO2/respiration) is noteworthy. The discussion could explore potential reasons for these discrepancies (e.g., device-specific limitations, participant compliance) and their implications for future studies.

Clarify whether the "126 gigabytes of compressed data" refers to raw or processed data, and how this volume translates into analyzable datasets.

Reviewer #4: this is a high quality pilot style project. The concept is interesting and may lead to a meaningful application of machine Learning in predicting the course and outcomes of THC use in PTSD and veterans.

The methods are sound and reasonable. The scales used such as (ISI,etc...) are appropriate.

The wearable devices are appropriate for such a study. The statistical methods are sound.

The results are reasonable and will help develop a larger scale study. Explaining what are potential differences between wearables (fitbit, apple, and if this may cause discrepancy in data ). Also more elaboration on the propriety data acquisition software is recommended.

I recommend explaining more clearly the ultimate goals of this study and how machine learning will be leveraged for this clinical challenge. Also briefly explaining to the readers what machine learning methods will be used and what strategy to optimize ML from this pilot.

**Do you want your identity to be public for this peer review?** For information about this choice, including consent withdrawal, please see our Privacy Policy

Reviewer #1: No

Reviewer #2: **Yes: ** OLAYINKA ADEBAJO

Reviewer #3: No

Reviewer #4: No

---

## [Decision Letter · Decision Letter 1]

28 Aug 2025

A remote measurement study of PTSD and cannabis use among veterans: recruitment, retention, and data availability

PONE-D-25-14945R1

Dear Dr. Leightley,

We’re pleased to inform you that your manuscript has been judged scientifically suitable for publication and will be formally accepted for publication once it meets all outstanding technical requirements.

Kind regards,

Firas H Kobeissy, PhD

Academic Editor

PLOS ONE

Additional Editor Comments (optional):

Reviewers' comments:

Reviewer's Responses to Questions

**Comments to the Author**

Reviewer #4: All comments have been addressed

2. Is the manuscript technically sound, and do the data support the conclusions?

Reviewer #4: Yes

3. Has the statistical analysis been performed appropriately and rigorously?

Reviewer #4: Yes

4. Have the authors made all data underlying the findings in their manuscript fully available?

Reviewer #4: Yes

5. Is the manuscript presented in an intelligible fashion and written in standard English?

Reviewer #4: Yes

Reviewer #4: After reviewing the authors responses, all comments addressed adequately, good luck with the longterm project.

**Do you want your identity to be public for this peer review?** For information about this choice, including consent withdrawal, please see our Privacy Policy

Reviewer #4: No

---

## [Editor Report · Acceptance letter]

PONE-D-25-14945R1

PLOS ONE

Dear Dr. Leightley,

I'm pleased to inform you that your manuscript has been deemed suitable for publication in PLOS ONE. Congratulations! Your manuscript is now being handed over to our production team.

Kind regards,

on behalf of

Dr. Firas H Kobeissy

Academic Editor

PLOS ONE